# Elastic analysis bridges structure and dynamics of an AAA+ molecular motor

**Victor Hugo Mello**[1], **Jiri Wald**[2,3,4], **Thomas C Marlovits**[2,3,4], **Pablo Sartori**[1]*

**1** Gulbenkian Institute for Molecular Medicine (GIMM), Lisbon, Portugal, **2** University Medical Center Hamburg-Eppendorf (UKE), Institute of Microbial and Molecular Sciences, Hamburg, Germany, **3** Centre for Structural Systems Biology (CSSB), Hamburg, Germany, **4** Deutsches Elektronen-Synchrotron Zentrum (DESY), Hamburg, Germany

* pablo.sartori@gimm.pt

**Data availability statement:** All relevant data are within the manuscript and its Supporting Information files.

## Abstract

Proteins carry out cellular functions by changing their structure among a few conformations, each characterised by a different energy level. Therefore, structural changes, energy transformations, and protein function are intimately related. Despite its central importance, this relationship remains elusive. For example, while many hexameric ATPase motors are known to function using a hand-over-hand alternation of subunits, how energy transduction throughout the assembly's structure drives the hand-over-hand mechanism is not known. In this work, we unravel the energetic basis of hand-over-hand in a model AAA+ motor, RuvB. To do so, we develop a general method to compute the residue-scale elastic pseudoenergy due to structure changes and apply it to RuvB structures, recently resolved through cryo-EM. This allows us to quantify how progression through RuvB's mechanochemical cycle translates into residue-scale energy transduction. In particular, we find that DNA binding is associated with overcoming a high energy barrier. This is possible through inter-subunit transmission of energy, and ultimately driven by nucleotide exchange. Furthermore, we show how this structure-inferred energetic quantification can be integrated into a non-equilibrium model of AAA+ assembly dynamics, consistent with single-molecule biophysics measurements. Overall, our work elucidates the energetic basis for the hand-over-hand mechanism in RuvB's cycle. Besides, it presents a generally applicable methodology for studying the energetics of conformational cycles in other proteins, allowing to quantitatively bridge data from structural biology and single-molecule biophysics.

## Author summary

Molecular motors are proteins that transform chemical energy into motion. To do so, they change their shape through a highly organised cycle. How exactly these changes in shape translate into energy transformations that ultimately result in motion is still not fully understood. In our work, we combined tools from structural biology and biophysics

**Funding:** This work was financially supported by the fellowship from Fundação para a Ciência e Tecnologia (UI/BD/152254/2021) to V.H.M., a laCaixa grant (LCF/BQ/PI21/11830032) to P.S., and by funds available to T.C.M. through the Behörde für Wissenschaft, Forschung und Gleichstellung of the city of Hamburg at the Institute of Microbial and Molecular Sciences at the University Medical Center Hamburg-Eppendorf (UKE) and Deutsches Elektronen Synchrotron (DESY). The funders had no role in study design, data collection and analysis, decision to publish, or preparation of the manuscript.

**Competing interests:** The authors have declared that no competing interests exists.

to better quantify and model this process. We focused on RuvB, a motor made of six identical parts that work together to move DNA through its central pore. We first tracked how RuvB's shape changes and how much energy these changes involve, and then built a theoretical model that simulates its operational cycle. More broadly, our approach shows how studying protein energetics can connect the perspectives of structural biology and single-molecule biophysics, thus providing new ways to understand how molecular machines work.

## 1. Introduction

AAA+ is a superfamily of protein assemblies that catalyse a wide range of cellular functions, including protein degradation and DNA recombination. These motors typically operate by translocating a polymeric substrate through the central pore of the assembly, a process driven by the ATP hydrolysis cycle [1]. Recent cryo-EM structures of hexameric AAA+ have consistently shown that they adopt an asymmetric assembly architecture, where individual subunits organise as a staircase around the translocated substrate [2–14]. Based on this structural insight, it was suggested that subunits coordinate to move relative to the substrate by sequentially gripping and releasing it in a *hand-over-hand* fashion [1]. However, the underlying basis for such orchestrated motion remains unclear.

From a thermodynamic perspective, hand-over-hand coordination arises from energy transduction. Energy transduction is fundamental to all molecular motors because it couples the energy released from a chemical cycle–ATP hydrolysis and nucleotide exchange–to the motion produced on a mechanical cycle–substrate translocation. Crucially, chemical energy is not only needed to sustain motion but also to ensure coordination of different components [15–19]. However, despite the central role of energy transduction in protein function [20–22], how this energy is distributed throughout the structure of proteins is largely unknown. Consequently, how to connect the energetics of protein function with detailed structural information remains an open question. This paper constitutes a step towards characterising the energetic basis of the hand-over-hand mechanism in an AAA+ motor at the structural level, thus connecting structural biology with protein biophysics for this paradigmatic example.

To achieve this goal, we build on previous work [23,24] to define a measure of structural deformation at a residue-scale, the elastic pseudoenergy $\Delta E_{el}$ (Fig 1A and 1B). While elasticity was originally aimed to describe deformations in large objects, proteins have been shown to exhibit elastic behaviour [25–31]. Furthermore, deformations such as bending [32,33], stretching [34], and twisting [35] are routinely (yet informally) ascribed to protein structures. Here, we provide a method (see *Methods* for details) to locally quantify a proxy for elastic energy from structural data, built on our recently published Protein Strain Analysis (PSA) python package [24]. This approach assumes proteins are homogeneous, isotropic elastic materials–a strong simplification that motivates our use of the term *pseudoenergy*. In the discussion, we detail the limitations and caveats of such method as well as possible improvements. Despite these shortcomings, $\Delta E_{el}$ correlates with metrics routinely used as proxies of deformation and is consistent with energy differences between haemoglobin conformations (see Appendix A in S1 Text), supporting its utility as a descriptive measure. We therefore use the elastic pseudoenergy as a statistical tool to identify candidate regions of high deformation. With these regions and the knowledge of the chemical cycle, we provide qualitative insights on energy transduction for a molecular motor in its operational cycle, Fig 1C.

We apply the framework described above to study how energy transduction promotes hand-over-hand in RuvB, an AAA+ hexameric assembly. RuvB is a molecular motor that

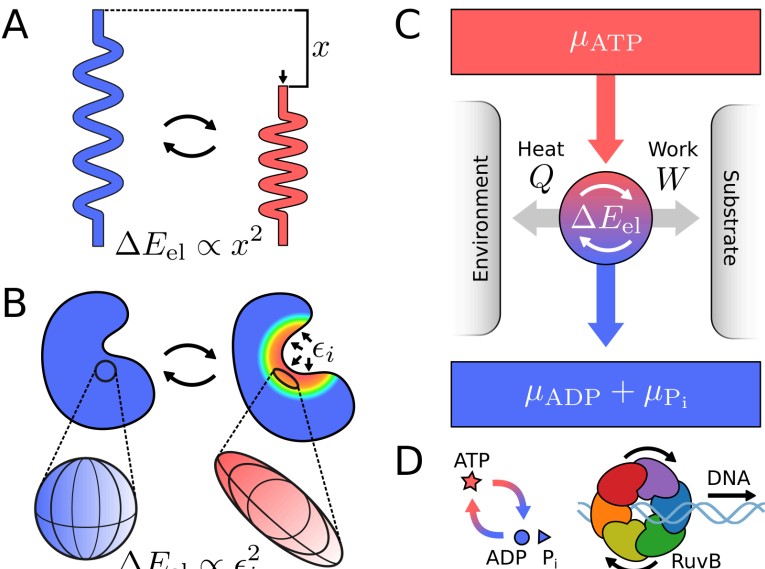

**Fig 1. Elastic deformations mediate energy transduction. A.** The spring is the simplest example of an elastic system. Hooke's law establishes that when a relaxed spring is compressed by a length $x$, the change in elastic energy behaves as $\Delta E_{el} \propto x^2$. **B.** For three-dimensional objects, like proteins, Hooke's law generalizes to continuum mechanics. A small region of a protein, depicted as a sphere in the reference conformation, deforms into an ellipsoid. This deformation is quantified by the local principal strains $\epsilon_i$ (with $i = 1, 2, 3$). The associated elastic energy change behaves as $\Delta E_{el} \propto \epsilon_i^2$. **C.** Energy transduction through a molecular motor occurs when the difference in free-energy between ATP ($\mu_{ATP}$), and ADP and $P_i$ ($\mu_{ADP} + \mu_{P_i}$) drives cyclic deformations in the motor ($\Delta E_{el}$), resulting in work performance and heat release. **D.** RuvB is a hexameric AAA+ molecular motor that transduces chemical energy into the motion of a DNA molecule through its central pore.

translocates DNA in the Holliday Junction, hence driving branch migration in bacteria, Fig 1D. We base our analysis on RuvB structures engaged in its mechanochemical cycle, recently visualized by time-resolved cryo-EM [2]. We first show that a simple principle of minimal elastic deformation can sort the conformational cycle of RuvB, in agreement with the result of manual sorting. We then identify the spatial distribution of elastic pseudoenergy along the conformational cycle. With this information, we propose that two distinct steps of the hand-over-hand cycle have specific energetic requirements. The *hand out*, or detachment from DNA, does not involve significant changes in elastic pseudoenergy, whereas the DNA *hand in* is associated with overcoming a large barrier. We suggest that the energy needed to overcome this barrier comes from nucleotide exchange, and is transmitted across assembly monomers. Finally, we integrate the structurally inferred pseudoenergy landscape into a non-equilibrium kinetic model of RuvB hexamers, which recapitulates our findings and predicts assembly dynamics. Overall, our work provides an energetic basis for hand-over-hand in RuvB. More generally, our method can be used to connect structural variability with energetics in other proteins, particularly those for which a series of conformations is captured "in action" using time-resolved cryo-EM.

## 2. Results

**Cycle reconstruction using minimal deformation principle.** To describe energy transduction of a protein at the structural level it is necessary to resolve the multiple conformations it adopts along its mechanochemical cycle. These "structural snapshots" can be inferred,

for example, from structural variability in the cryo-EM ensemble [36]. Once the conformations are obtained, a key task consists of finding their correct time order in the cycle, typically through visual inspection. Can an elasticity-based approach provide a systematic solution to this task?

For the case of the RuvB homohexamer, we consider five structures of the assembly $(s_1, \ldots, s_5)$ and a putative reference not engaged on the cycle $s_0$, which were obtained and manually ordered in [2], Fig 2A and *Methods*. Upon the completion of one cycle, each of the RuvB subunits switches to the next position relative to the DNA molecule. Altogether, the five cycle hexamers and the position switching rule imply that RuvB subunits perform a cycle with 30 conformations, Fig 2B. From now on, we refer to each of these conformations by a letter, indicating a position relative to the DNA (with A in the top of the staircase, followed by B, C, …), and a number, indicating the numbering of the structure, e.g., F1, F2, …, A5. Crucially, ordering the five structural snapshots as done in [2] involved extensive and time-consuming geometrical analysis, visual inspection, and careful evaluation of nucleotide-binding sites' electronic densities. We now provide an alternative approach for time ordering based on protein elasticity.

Specifically, we test whether a principle of minimal elastic deformation –by which subsequent structures in the cycle are minimally deformed– can recapitulate the order of these snapshots. To do so, we first calculate the residue-scale strain tensor–a quantity foundational to elasticity theory that measures local deformation [37]–over all possible pairs of conformations. Second, from this tensor we extract the local principal strain $\epsilon_3$, compute its average over all residues, and construct a matrix of size $30 \times 30$ with all pairwise comparisons (see S1 Fig for an analysis including $s_0$). Pairs of conformations to which correspond a large matrix element are very different, as they relate to each other by a large deformation. Third, to interpret the matrix of deformations, we project it in two dimensions using Multidimensional Scaling (MDS) [38,39]. The resulting projection shows that subunits cluster in agreement with their position in the assembly (different colors), Fig 2C. Furthermore, within these clusters, we can identify the order of the cycle substeps (1 to 5), which corresponds to the temporal order of the hexameric assemblies (see Appendix C in S1 Text for a detailed description and comparison with other approaches). Overall, our unbiased approach independently validates the ordering of structural snapshots in the mechanochemical cycle of RuvB.

**Elastic pseudoenergy of the mechanochemical cycle at the subunit and residue scales.**
Having shown the importance of elasticity in recovering the RuvB cycle, we now compute the elastic pseudoenergy of all thirty conformations. To do so, we use as reference an assembly structure, $s_0$, that is not engaged in the mechanochemical cycle [2] and compute the elastic pseudoenergy of each subunit compared to the reference, see *Methods* for details and Appendix D in S1 Text for other reference choices. Fig 3A shows the magnitude of elastic pseudoenergy as a heat map on each conformation. We find that the spatial distribution of elastic pseudoenergy is highly heterogeneous, with small regions accumulating most of the pseudoenergy. Additionally, this distribution exhibits significant variation across different conformations (S2 Fig). We now quantitatively analyse these energetic patterns.

To obtain a global view of RuvB energetics we compute the total elastic pseudoenergy per subunit, Fig 3B. The resulting pseudoenergy landscape is smooth and consists of a steady build-up of an energetic barrier in DNA free states and a decrease to a flat landscape in DNA bound states. Note that the highly energetic conformations (D, E and F) correspond to the "converter module" hypothesised in [2] to mediate energy transduction. At a local scale, we describe the elastic pseudoenergy for each residue by a trajectory over conformations, Fig 3D. These trajectories show that different residues are more deformed in specific states of the mechanochemical cycle. We identify four mechanically active regions based on the variability

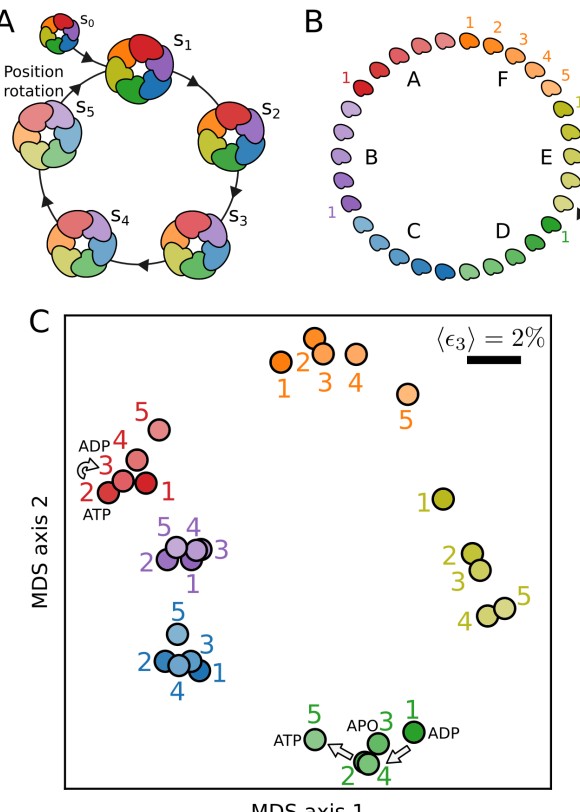

**Fig 2. Minimal deformation principle reconstructs RuvB monomer cycle. A.** Sketch of RuvB hexameric cycle (states $s_1, \ldots, s_5$) together with the initial state ($s_0$). In the final transition, $s_5 \rightarrow s_1$, subunits switch to the next position. **B.** The ordered assembly states together with the position switching rule imply that each RuvB subunit undergoes a cycle of 30 conformations. **C.** Low-dimensional embedding of pairwise comparisons between RuvB subunits is arranged in a cycle. Conformations cluster according to their position and a parsimonious order from 1 to 5 can be deduced for positions F, A, E, and D. We use Multidimensional Scaling on the mean strain values, $\langle \epsilon_3 \rangle$, for each pairwise comparison. The length of the scale bar indicates a mean strain of 2% for both axes. The plot depicts the mean of 100 individual MDS realisations.

of the pseudoenergy trajectories, Fig 3C and 3E (see *Methods*). Interestingly, all four regions are discontinuous in sequence but perform similar interaction patterns (S3 Fig and S1-S4 Videos), suggesting that despite containing relatively distant residues, each region is consistently related to the same function (note that the same was not observed for $\beta$-factors as a metric instead of pseudoenergy, S4 Fig).

Region 1 comprises part of the nucleotide-binding pocket and connects the small and large ATPase domains. Region 2 localises mostly to sensor 1, a loop motif conserved in all AAA+ [40] whose specific role is critical for function but not consensual among studies [1,41–43]. Region 3 belongs to the head domain (C-terminal), which interacts with the DNA substrate. Region 4 largely coincides with a conserved structural motif named pre-sensor I $\beta$-hairpin, an insertion common to PS1 clade of the AAA+ superfamily, crucial for function [44]. In the context of RuvB, this motif provides direct and transient protein-protein interactions between RuvB and RuvA [45].

To summarise, we have identified the pseudoenergy landscape of RuvB and key mechanically active regions in the structure. In the following, we study energy transduction through

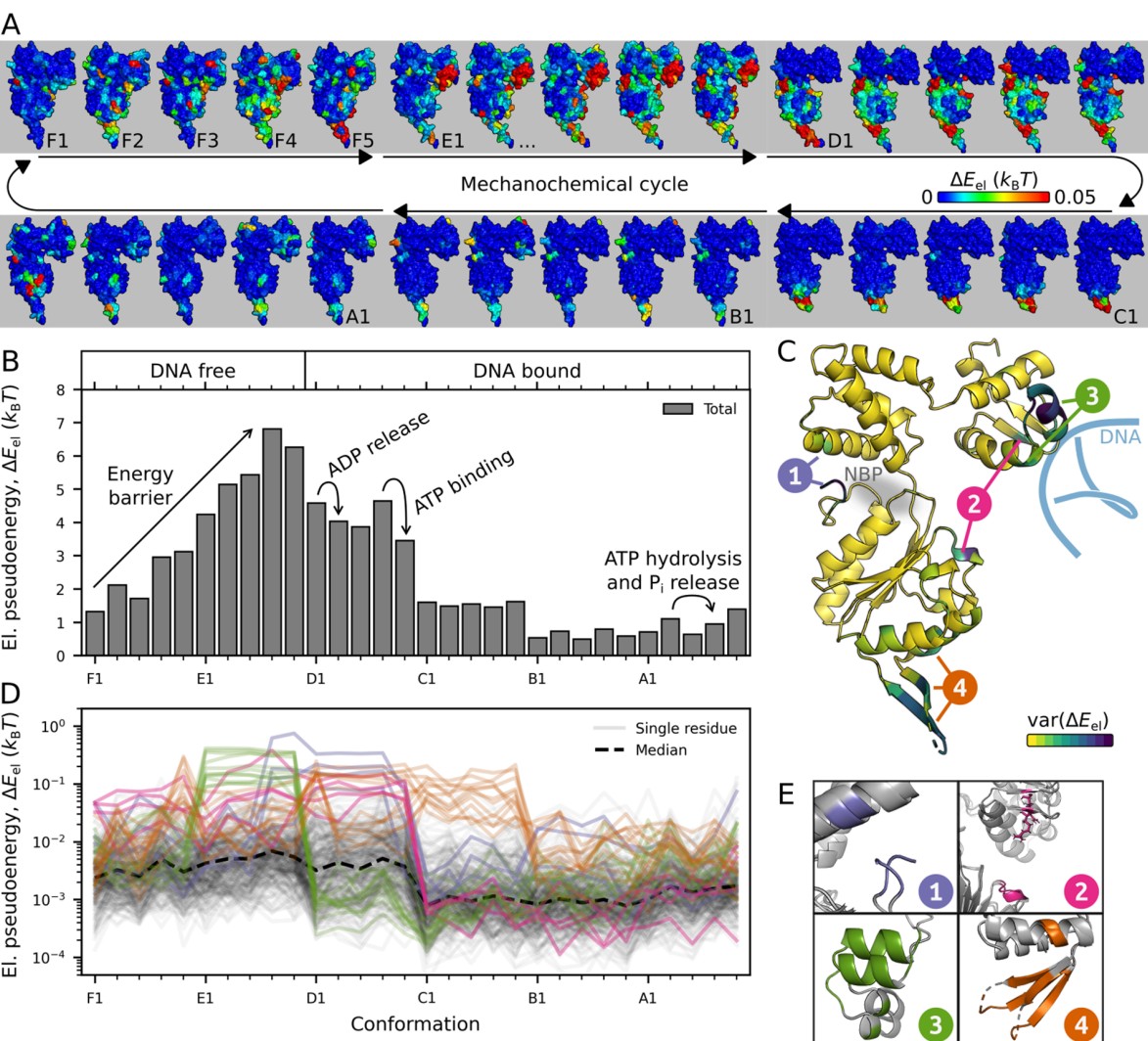

**Fig 3. Elastic pseudoenergy content of a RuvB subunit along its mechanochemical cycle. A.** We calculate the elastic pseudoenergy content in the RuvB's protein structure at a residue level for its conformational cycle in units of $k_BT$, with $k_B$ the Boltzmann constant and $T$ the temperature. **B.** The sum of all elastic pseudoenergy is represented throughout the cycle, representing RuvB's pseudoenergy landscape. On top, the conformations are labelled according to their nucleotide content and engagement or not with the substrate. **C.** RuvB subunit (A1 conformation in the figure) color-coded with the variance of elastic pseudoenergy across conformations. The most variable residues, grouped into four mechanically active regions, are marked by dark blue colors. Note that region 1 localises close to the nucleotide-binding pocket, and region 3 localises close to the DNA molecule. **D.** Elastic pseudoenergy trajectories of each residue throughout the conformational cycle. The top 10% residues with most energetic variability display different characteristic trends of elastic pseudoenergy trajectories, which led us to identify the mechanically active regions, shown in different colors. **E.** The structure of the four regions is displayed in two conformations with different energy contents. The regions are numbered and colored consistently throughout the manuscript.

these regions and its connection to hand-over-hand. To this end, we analyse energetic exchanges with the DNA substrate (region 3, related to translocation), across RuvB subunits (regions 2 and 3, related to inter-subunit coupling), and with the nucleotide (region 1, related to chemical driving). Deformations in region 4 relate to the dynamic interactions with RuvA, therefore playing a role in the assembly cohesion, which is outside the scope of this paper. Altogether, these elements compose the energetic contributions for the function of a molecular motor, Fig 1C.

**Energy is released upon transition to DNA-bound state.** A central aspect of molecular motor energetics is that they perform mechanical work by moving relative to a substrate [25]. For some AAA+, the motion of the substrate through the pore arises from a sequential hand-over-hand action of the different subunits [1], as is apparent by the staircase arrangement of the subunits around DNA, Fig 4A. To quantify the energetics of this translocation process, we identify all the residues in RuvB that establish polar interactions with DNA across each conformation, (see *Methods* for the procedure and S5 Fig for all polar interactions). The RuvB/DNA interface is highly dynamic and comprises six residues, some of which belong to regions 2 and 3. Because most of these interactions are established in conformation D1 and are lost in conformation A5, we name *hand-in* the transition E→D, and *hand-out* the transition A→F.

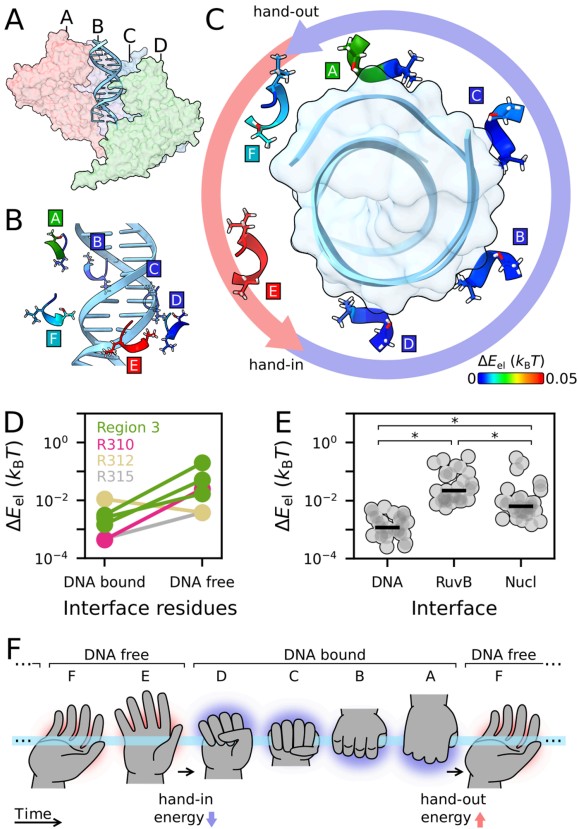

**Fig 4. Energetics of the RuvB/DNA interface. A.** Global view of the RuvB subunits A, B, C and D arranged as a staircase around the DNA (assembly state $s_5$). **B/C.** Residues G279-T282, belonging to region 3, colored by elastic pseudoenergy (side view in B and top view in C). **D.** Mean elastic pseudoenergy content of residues in DNA bound and DNA free states. Except for R312, all residues display higher elastic pseudoenergy when not interacting with the DNA, which corresponds mostly to positions F and E. **E.** Total elastic pseudoenergy of different interfaces, with each data point corresponding to a conformation. RuvB/DNA interface is less energetic than RuvB/RuvB and RuvB/Nucleotide interfaces throughout the mechanochemical cycle (Kruskall-Wallis test $p < 0.05$; Mann-Whitney U test, $p < 0.05$, Bonferroni corrected). **F.** Sketch representing "hand-over-hand" with the energy content of RuvB/DNA interface appearing as a red (high energy) or blue (low energy) glow. Each hand represents the same RuvB subunit progressing over time through its conformational cycle (from left to right). For visual clarity, only six of the thirty conformations are shown.

Fig 4B and 4C show the energy content of region 3 residues in the RuvB/DNA interface. Remarkably, we find that these residues interact with DNA when they are in a low pseudoenergy state, colored in blue (see also S6 Fig). Conversely, they exhibit high elastic pseudoenergy when they do not interact with the DNA, Fig 4D. When compared to other interactions, such as with the nucleotide and across RuvB subunits, interactions with DNA are characterised by a low elastic pseudoenergy, Fig 4E. Overall, this suggests that after DNA hand-out the interface accumulates energy that is then released upon DNA hand-in. Therefore, DNA stabilises the interface, see Fig 4F for schematics. Next, we investigate how inter-subunit interactions play a role in transmitting energy to this interface during the DNA free conformations, i.e. between *hand-out* and *hand-in*.

**Inter-subunit coupling drives DNA hand-in.** Above, we showed that in DNA free states RuvB increases its energy, i.e., it "climbs" an energy barrier. How can these energetically unfavourable dynamics unfold? To address this question, we now investigate energy transduction across neighbouring RuvB subunits by characterising their interface. As shown in Fig 5A, we define the forward subunit (pink) as the subunit forward to the interface according to the cycle orientation, and correspondingly for the backward subunit (gray). We find that regions 2 and 4 dominate the elastic pseudoenergy at the forward subunit (Fig 5B), whereas region 3 accounts for most of the elastic pseudoenergy at the backward subunit (Fig 5C). In particular, elastic pseudoenergy is high when the forward subunit is in the DNA bound position D and the backward subunit in the DNA free position E. Therefore, the interface between subunits D/E has increased elastic pseudoenergy on both sides.

We now focus on the interaction between regions 2 and 3, which come in contact with each other from opposite sides of the interface throughout the cycle (Figs 5D-F and S3 for the complete interaction profile). From the inspection of these regions at a molecular level, we observe two noticeable patterns: *(i)* in region 3 at the backward subunit, the highly energetic states correspond only to conformation D1 to D5, where a large displacement of the whole head domain is observed; and *(ii)* in region 2 at the forward subunit, the increase and decrease of elastic pseudoenergies correlate with switches in the secondary structure from a loop (high energy, F1 to D5) to a $3_{10}$ helix conformation (low elastic pseudoenergy, C1 to A5), see S7 Fig. Note that $3_{10}$ helices are sterically unfavourable [46,47], suggesting that folding can be involved in transient energy storage. In summary, these deformation patterns indicate that energy storage and release in region 2 resemble a spring-like mechanism, by which one subunit assists the displacement and repositioning of its neighbour. Such rearrangement accommodates RuvB within the double-helix minor groove, a contact that establishes the basis of hand-in for RuvB, see Fig 5G.

**Nucleotide exchange is the energy source of the mechanochemical cycle.** So far we have shown that the mechanochemical cycle of RuvB is characterised by an energy build-up just before DNA hand-in, and that this build-up is associated with inter-subunit energy coupling. What is the ultimate source of this energy?

Following the framework in Fig 1C, it is clear that the source of energy is in the nucleotide cycle, and thus we turn our attention to the nucleotide binding pocket (NBP). The NBP is near region 1, identified in Fig 3. This small region presents two different conformations, closed and open, acting as a gate, Fig 6A. In the open conformation, both ADP release and ATP uptake take place; in the closed conformation, hydrolysis takes place.

We find that region 1 displays a high pseudoenergy while in the open conformation, when it also interacts with the nucleotide, see Fig 6B. In fact, in its open conformation region 1 accounts for 60% of the elastic pseudoenergy of residues interacting with the nucleotide, largely surpassing the core of the NBP, which is mostly composed by the Walker A motif

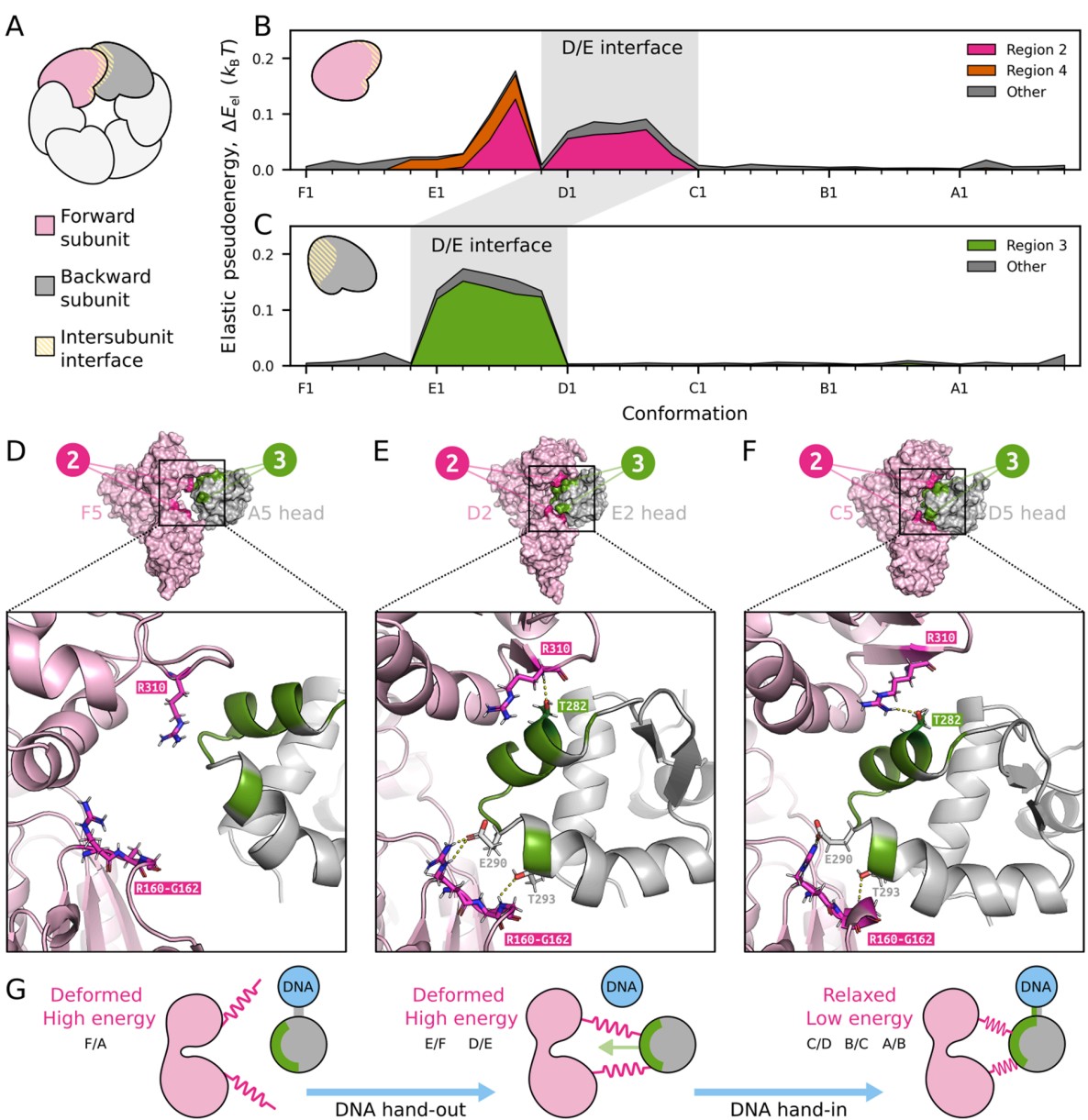

**Fig 5. Energy transduction across adjacent RuvB subunits. A.** Schematic of the interface between the forward and the backward sub-units. The nomenclature *backward* and *forward* refers to their order on the conformational cycle. **B/C.** Total elastic pseudoenergy at the interface on the forward subunit (panel B) and backward subunit (panel C). Regions 2 and 4 represent a large fraction of the total elastic pseudoenergy of the former and region 3 of the latter. **D/E/F.** Representative structures of the interaface dynamics: F5/A5 in panel D, where no interactions are found across subunits; D2/E2 in panel E, where these regions display polar interactions and residues R160-G162 adopt a loop structure; and C5/D5 in panel F, where regions 2 and 3 interact, but R160-G162 adopt a $3_{10}$ helix structure. **G.** Schematic of the interaction and energy patterns between neighbouring RuvB monomers: forward subunit is in a high-energy state, it then approaches and binds the backward subunit, finally it releases energy as the backward subunit moves and binds to DNA.

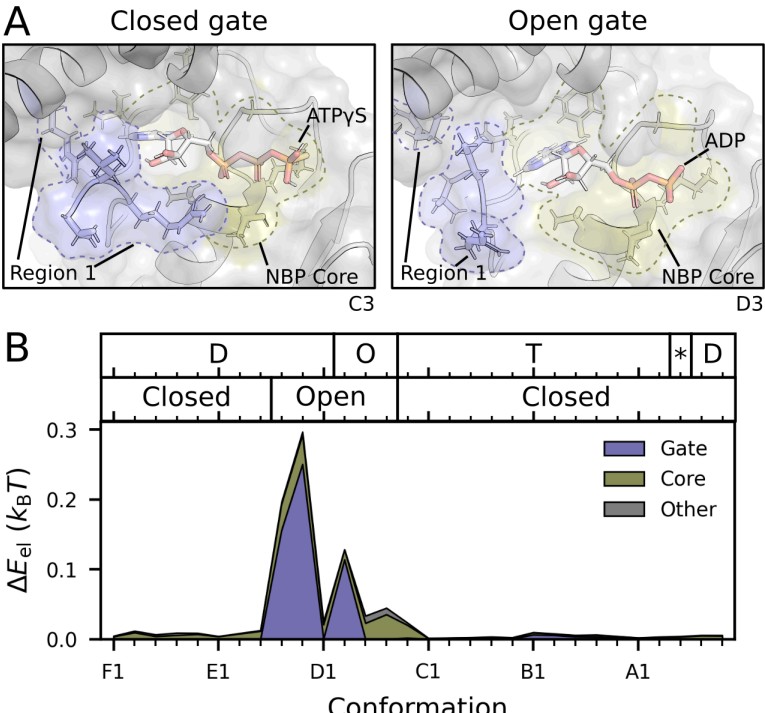

**Fig 6. Nucleotide exchange drives hand-over-hand. A.** Region 1, which surrounds the NBP, adopts two distinct conformations: closed, when the nucleotide is ATPγS (a slowly hydrolysable ATP analogue); and open, when the nucleotide is ADP. **B.** The elastic pseudoenergy of residues interacting with the nucleotide is high and dominated by region 1 when this region adopts its open conformation. Top labels indicate nucleotide state in each conformation (D for ADP, O for empty, T for ATP, and ∗ for ADP + $P_i$).

residues (see S8 Fig for an analysis of Walker A and other conserved residues). In contrast, region 1 has low pseudoenergy in its closed conformation.

This implies that the imbalance in chemical potential, which fuels RuvB dynamics, has a direct consequence on structural deformations during nucleotide exchange, but not hydrolysis. Furthermore, the open conformation of region 1 co-occurs with the conformations in which the interface between subunits is highly energetic, Fig 5. Taken together, these observations suggest that the free-energy of ADP binding is transduced across subunits to mediate the highly energetic hand-in interaction. This completes the energy transduction cycle behind hand-over-hand and substrate repositioning.

**Kinetic model of mechanochemical cycle constrained by structural analysis.** So far, we have studied the basis for the hand-over-hand mechanism in RuvB using structural data analysis. Building on this, we now propose a minimal kinetic model for RuvB dynamics. This model relies not only on the intuition learned from the structural analysis but also on the total pseudoenergy for each state, which we use to constrain energetic changes along the cycle. While obtaining dynamic information from structural snapshots is clearly impossible, such energetic constraints allow us to test whether our previous observations are thermodynamically consistent with processive motion. Specifically, we ask whether RuvB subunits can overcome a $\approx 6k_BT$ energy barrier relying on intersubunit coupling. Next, we introduce the main elements of the toy model inspired by our previous analyses.

The dynamics of motor proteins can be modelled as stochastic transitions among a small set of states, see e.g. the classic works [20,25,48–50]. In the case of RuvB, an assembly state

corresponds to each subunit adopting one of the 30 conformations described in Fig 2B. The energetics of the model incorporates three central concepts discussed so far: a potential energy per conformation, which we identify as the elastic pseudoenergy $\Delta E_{el}$; energy due to inter-subunit coupling, $E_c$; and free-energy exchange due to the nucleotide cycle, $\Delta\mu$ (Fig 7A and *Methods* for mathematical formulation). Finally, we assume that all transitions between conformations conserve energy, except those that dissipate energy from the nucleotide cycle (see S10 Fig). This allows us to simulate the dynamics of an assembly quantitatively constrained by our previous energetic analysis of structures, Fig 7B. To adjust the time axis $t$, we calibrate the characteristic time scales of conformational changes using published experimental data on branch migration assays catalysed by RuvAB [51], see details in Appendix F in S1 Text.

We next examine the impact of altering the order of the energy landscape in the conformational cycle, see Fig 7C. Two ensembles of randomised conformational landscapes were generated: one in which the elastic pseudoenergies of the thirty conformations were fully shuffled (gray), and another in which only the order of the assemblies ($s_1$ to $s_5$) was shuffled (magenta), while maintaining the subunit order (F to A). The resulting speed distributions reveal that the native order of conformations (blue) yields significantly higher processive speeds compared to randomised cycles. This outcome is intuitive, as the native elastic pseudoenergy landscape is relatively smooth, facilitating efficient progression. In contrast, shuffled potentials are generally more rugged, increasing the likelihood of RuvB subunits to dwell

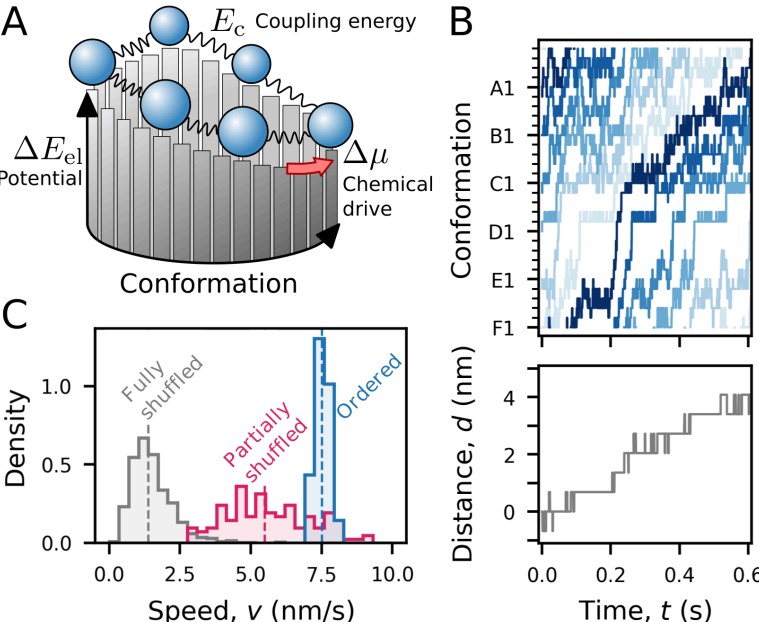

**Fig 7. A kinetic model links protein energetics to protein dynamics. A.** Sketch of the kinetic model. The physical system comprises six subunits that populate conformations in a periodic mechanochemical cycle. Each conformation is associated with a potential energy, which we associate with $\Delta E_{el}$ (bars on the vertical axis). Adjacent subunits interact through a coupling energy $E_c$ (represented by springs). Some transitions are biased in a particular direction (red arrow), with a chemical drive corresponding to the free-energy from the nucleotide cycle $\Delta\mu$. **B.** Trajectory of RuvB conformations simulated with the model (top, subunits in shades of blue) and translocated distance of the DNA molecule (bottom, in gray). **C.** Distribution of speeds simulated from the model with shuffled and ordered elastic pseudoenergy landscapes. The sample sizes are $n = 1000$ (fully shuffled); $n = 120$ (partially shuffled); and $n = 20$ random seeds (ordered).

in local energy minima. Overall, these results suggest that our model, constrained by structures, results in robust assembly dynamics (see also Appendix E in S1 Text for an analysis of the impact of each energetic contribution on the translocation speed).

**Simulated model trajectories recapitulate structural observations.** With the model parametrised, we can now computationally investigate the collective dynamics of the RuvB assembly. In particular, we focus on characterising the inter-subunit coupling in order to evaluate whether the model recapitulates our structural observations.

Our method inherently considers that adjacent subunits are allowed to transfer energy, but it does not set a priori the magnitude of this coupling at each conformation. To quantify the magnitude of the inter-subunit coupling throughout the cycle, we compute the coupling energy between adjacent subunits, see Appendix G in S1 Text. Fig 8A shows that forward and backward coupling energies are localised to conformations before and after the largest energy barrier. This supports that inter-subunit coupling is essential for cycle progression, which is mainly hindered by a large energetic barrier in position E.

We then calculate the mean coupling force across subunits in each cycle conformation. The force, which corresponds to the spatial derivative of the coupling energy (Appendix G in S1 Text), determines the direction and intensity with which subunits push or pull each other. Fig 8B shows that the mean resulting force is positive (aligned to the cycle) in positions F, E, and D, and negative (opposed to the cycle) otherwise. Note that the assisting role of positive forces in crossing the energy barrier is reminiscent of the previously proposed role of the converter module [2,52]. Overall, the emergent energy coupling landscape predicted by the model supports the idea that coupling is key in overcoming the hand-in energetic barrier.

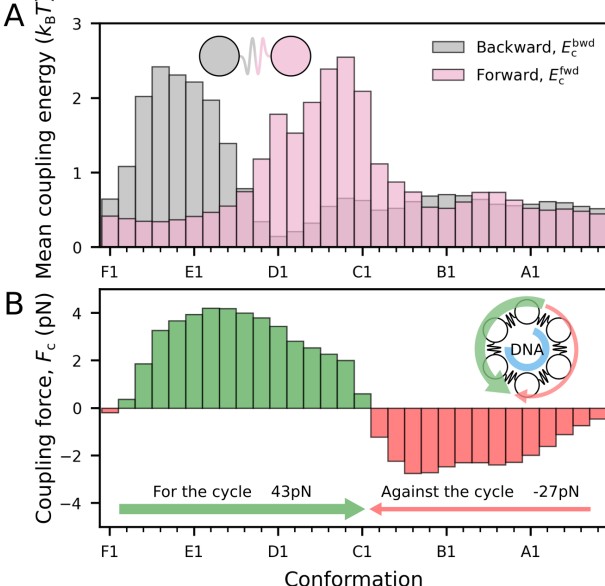

**Fig 8. Coupling energy prediction of kinetic model. A.** Mean coupling energy at backward and forward interfaces (see schematic). The coupling energy is high before and after the energy barrier. **B.** Total coupling forces, summed effect of the spatial derivative of backward and forward coupling energies. Bars in green represent forces aligned to the cycle direction (F → E) and in red, in the opposite direction (E → F). The model predicts a net positive force over the whole cycle. The cartoon represents the RuvB hexamer, with green and red curved arrows for the positive and negative forces, respectively.

## 3. Discussion

**Summary.** Protein function depends on structural deformations. Despite much of these deformations being caused by spontaneous thermal fluctuations, coordinated motion requires energy input [53]. This is particularly notable for motors, such as those in the AAA+ superfamily, in which function is directly driven by the free-energy released from the nucleotide cycle. Despite this intimate relationship, protein energetics and function remain largely disconnected at the structural scale. In this work, we have linked these two aspects of proteins for a particular AAA+ assembly, RuvB. We have done so by following an interdisciplinary approach that combines elasticity-based data analysis of experimentally determined structures [23,24,54–58], and non-equilibrium thermodynamics modelling [20,25,48–50]. The key contributions of our work are twofold: a general methodology applicable to any ensemble of structures, and novel energetic insight into AAA+ hand-over-hand mechanism.

**Methodological considerations and caveats.** Our approach builds on the previously established Protein Strain Analysis [23,24] and extends it with two new applications: first, unsupervised ordering of an ensemble of structures using strain as a dissimilarity metric; second, obtaining residue-scale energetic information that is then integrated in a minimal mathematical model. These applications are completely general and can be deployed to any protein for which multiple structures have been experimentally or computationally resolved. Nevertheless, the success of these approaches depends on how accurately the conformational ensemble samples the underlying protein dynamics.

Concerning the ordering of structural ensembles, there are related methods that investigate conformational variability. Examples are methods relying on Ensemble PCA or NMA [59–62], as well as methods aimed at heterogenous cryo-EM reconstructions [36,63,64]. Compared to the first group, our approach analyses dissimilarities between pairs of structures, instead of structure-specific features such as atomic coordinates (for ensemble PCA) or their vibrational modes (for ensemble NMA). An additional difference is that PSA is independent of structural alignment, thus better characterising motions between rigid domains [23,24]. Compared to the second group, our approach presents the key difference that it relies on atomic maps, not raw EM micrographs. On the one hand, this makes PSA suitable to assess structures regardless of their acquisition approach (cryo-EM, X-ray diffraction, NMR, or computational structure prediction). On the other hand, our method is sensitive to the accuracy of the atomic-map reconstructions, thus requiring quality control assessments before their application to PSA. A natural next step is to adapt the elastic approach to process raw experimental data, e.g. electron densities. Overall, our method provides a readily interpretable low-dimensional representation of a set of structures.

Concerning energetics of protein structures, previous approaches have focused on the construction of energy landscapes from the raw ensemble of single-molecule cryo-EM images [63,65–67] and molecular dynamics [68]. The key distinctive feature of our method is that it provides energetic information at the residue-scale. To achieve this, we relied on a simple yet strong assumption: that the deformation of a protein relative to a reference state is well captured by a homogeneous and isotropic elastic model. While proteins have long been ascribed elastic properties [28,69,70], our model assumptions are a clear oversimplification. For example, proteins are generically anisotropic and non-homogeneous materials, as they deform more easily in specific directions and on specific regions. Despite this critical disadvantage, we show in Appendix A in S1 Text that our proposed metric correlates significantly with other measures (and proxies) of energy or deformation at atomic scales, while matching the order of magnitude of energy differences at a mesoscopic scale, typical of biophysical studies.

A natural extension of our method is to incorporate Elastic Network Models (ENM) to estimate protein stiffness. In ENM, atoms in a single structure and within a cutoff distance are connected by springs, with coupling strengths inferred from $\beta$-factors [71]. While this framework does not directly translate to 3D continuum mechanics, it can inform analogous stiffness calibration strategies for PSA. The same idea applies to ENM variants that account for heterogeneity [72] and anisotropy [73]. Also, reference structures for deformations are likely pre-stressed [74,75], which implies that simplistic elastic models such as the one considered here may under or overestimate the energy of a deformed conformation. A potential way of addressing this is to consider plastic deformations in addition to the standard elasticity theory as a framework to model proteins [76]. Finally, it is possible that further experimental advances in single-molecule mechanics can provide more detailed local information on protein stiffness.

**Relationship between mechanical and biochemical activity.** Our approach identifies few and small protein regions where deformations are large. A natural question is whether mechanically active regions are also likely to display specific biochemical functions.

In S8 Fig we show that there is no clear relationship between high elastic pseudoenergy and sequence conservation, which we use as a proxy for conservation of biochemical properties. In particular, in the mechanically active regions we find residues ranging from strongly conserved to not conserved at all. This might be due to a small overlap with biochemically active residues, which exhibit a large degree of conservation, whereas the remaining residues deform because they are contiguous in sequence. It is possible that this deformation has a mechanical role in transiently storing energy through deformation, and therefore does not require side chains with specific biochemical properties. Just as biochemical activity is associated with sequence conservation, it remains open whether mechanical activity is also associated with conserved structural features. This also raises the question of whether conservation at the motif level, rather than the residue level, provides a better match to mechanical activity. We envision that such hypotheses can be experimentally tested using nano-rheological viscoelastic measurements [28,31], which have tremendous potential to disentangle the complex relationships among function, mechanics, biochemistry, and conservation.

**Biological aspects of hand-over-hand in RuvB.** In oligomeric AAA+ ATPases, hand-over-hand establishes that each subunit within the assembly sequentially alternates its conformation as it interacts with a substrate. Through a series of structural snapshots depicting actively processing RuvB in the RuvAB-Holliday Junction complex, the hand-over-hand mechanism was observed for RuvB [2]. After each cycle, each RuvB subunit adopts the conformation of its preceding neighbour. These conformations, characterised by their helical positioning relative to the DNA, play distinct roles in the DNA hand-over-hand process.

Our energetic analysis provides insights into these roles and how they interrelate. We found that the progression of the mechanochemical cycle is associated with overcoming an energy barrier during DNA-free conformations (F and E). Our findings suggest that mechanical coupling between subunits D and E plays a crucial role in assisting with the crossing of this barrier. Furthermore, we found that elastic deformations in the NBP during nucleotide exchange, rather than ATP hydrolysis, supply the energy that is transduced through inter-subunit coupling. This is in line with observations in other molecular motors, see [77–79]. In conclusion, an energetic cascade that starts in the NBP and is transmitted across chains leads to DNA hand-in, and constitutes the energetic basis of hand-over-hand in RuvB.

**Perspective.** While the hand-over-hand mechanism is widely observed among AAA+, our work is the first to provide an energy-based proposal of how it is coordinated. The approach we developed is easily translatable not only to other AAA+ that operate via hand-over-hand but also to those relying on different mechanisms. A few examples of these are the protease

ClpXP, which shows evidence of both stochastic ATP hydrolysis and sequential bursts [80]; and the bacterial pilus motors PilB and PilT-4, which exhibit two-fold symmetric structures during their functional cycles [81]. Overall, the generality of our approach, which combines physics-based structural analysis with minimal mathematical modelling, paves the way to a systematic investigation of the interplay between function and energy transduction in AAA+.

## 4. Methods

**A primer in Protein Strain Analysis.** To estimate the elastic pseudoenergy associated with protein conformational changes, we first need to introduce strain, a well-established physical concept that quantifies deformation. In order to do so, we briefly describe the mathematical framework of elasticity theory. This method is explained in detail in [24] and implemented in the PSA python package available at github.com/Sartori-Lab/PSA.

Consider the three-dimensional coordinates of a given atom $i$ of a protein in two conformations, $\mathbf{x}_i$ and $\mathbf{X}_i$. A local deformation between these conformations is defined by the motion of the atom $i$ relative to its surrounding atoms, and it is quantified by the deformation gradient tensor $\mathbf{F}_i$. This matrix can be estimated using the atoms in a neighbourhood of the given atom $i$ as

$$\Delta \boldsymbol{x}_i^j \approx \mathbf{F}_i \Delta \boldsymbol{X}_i^j, \tag{1}$$

where $j$ are all atoms in a local neighbourhood of $i$, and $\Delta \boldsymbol{x}_i^j$ and $\Delta \boldsymbol{X}_i^j$ are the displacement vectors between the atoms $i$ and $j$ in the different conformations. See Appendices B and C for a formal definition of the neighbourhood used in this study. The above equation is typically over-determined, and we thus estimate $\mathbf{F}_i$ minimizing a quadratic error as proposed in [82], see also [24]. From the deformation gradient, we define the Lagrangian strain tensor $\mathbf{E}_i$, which captures the finite deformations around the atom $i$ regardless of coordinate system and has the advantage of being independent of structural alignment:

$$\mathbf{E}_i = \frac{1}{2} \left( \mathbf{F}_i^{\mathrm{T}} \mathbf{F}_i - \mathbf{I} \right). \tag{2}$$

The strain tensor can be viewed as compressions or expansions in three orthogonal directions. This is represented schematically in Fig 1B by the deformations that transform a sphere into an ellipsoid. Associated to this view, we define the three principal strains $\epsilon_{i,a}$, with $a = 1, 2, 3$, the eigenvalues of the strain tensor given by $\mathbf{E}_i \mathbf{v}_{i,a} = \epsilon_{i,a} \mathbf{v}_{i,a}$, where the eigenvectors $\mathbf{v}_{i,a}$ are the directions of each principal strain. The principal strains are ordered as $\epsilon_{i,1} < \epsilon_{i,2} < \epsilon_{i,3}$.

**Elastic pseudoenergy from protein structures.** In order to discuss protein energetics, we use a hyperelastic model for the energy density, $\Psi(\mathbf{E})$ [83]. Specifically, we use a Saint Venant-Kirchhoff model for which $\Psi$ is given by:

$$\Psi(\mathbf{E}_i) = \frac{\lambda}{2} \mathrm{tr}(\mathbf{E}_i)^2 + \mu \mathrm{tr}(\mathbf{E}_i^2), \tag{3}$$

where $\lambda$ and $\mu$ are the first and second Lamé constants, respectively. The advantage of using this form of $\Psi$ is that $\lambda$ and $\mu$ relate to the Young modulus, which has been measured experimentally and estimated computationally for proteins [25,69,70,84–86]. This is not the case for most hyperelastic models. Also, this form of $\Psi$ is a strictly non-negative function, unlike other simple elastic models that might yield negative energies for compressed regions.

Because energy is an extensive quantity, it is defined as the integral of the energy density over the protein's volume. We compute the elastic pseudoenergy $\Delta E_{\mathrm{el}}$ contained in the

deformations of the neighbourhood of an atom as

$$\Delta E_{\text{el},i} = \Psi(\mathbf{E}_i)\nu_i\phi,$$
$$\phi = \frac{V}{\sum_{j=1}^{N} \nu_j},$$

(4)

where $\nu_i$ represents the van der Waals volume of the atom $i$, and $\phi$ is a scaling factor that depends on the van der Waals volume of the whole protein $V$ and the sum of volumes of the $N$ atoms considered in the analysis. The scaling factor $\phi$ is especially useful for adjusting the elastic pseudoenergy definition to different choices of atoms (only $C_\alpha$, main chain, all atoms, etc).

**Structural data.** The cryo-EM structures were downloaded from the Protein Data Bank (PDB) and are designated throughout the text as $s_0$: 7PBR, $s_1$: 7PBL, $s_2$: 7PBM, $s_3$: 7PBN, $s_4$: 7PBO, and $s_5$: 7PBP, following the nomenclature used in [2]. The RuvB subunits are labelled according to the original PDB file annotations, where chain A represents the top of the staircase interacting with the substrate, followed by chains B, C, and D as substrate-engaged, and chains E and F as substrate-disengaged. Additionally, residues not modelled in at least one conformation were excluded from all subsequent analyses.

**Cycle reconstruction.** In brief, to establish the relationship between the 30 RuvB conformations, we use PSA to compare each of them pair-by-pair, resulting in a dissimilarity matrix of $30 \times 30$ elements. We then further reduce the dimensionality of this matrix using the algorithm of multi-dimensional scaling (MDS). See the Appendix C in S1 Text for a more detailed description.

**Elastic characterisation of RuvB mechanochemical cycle.** To perform the elastic pseudoenergy calculations, we used PSA considering $s_0$ as the reference state for each pairwise comparison. Thus, each cycle conformation is compared to its cognate in the reference structure, e.g., the reference for conformation F1 is F0, for F2 is F0, and so on.

Additionally, we used in PSA a neighbourhood selection method that establishes a common list of neighbouring atoms regardless of the structural comparison. For a complete description of the parameters we used, see Appendix G in S1 Text. We also evaluated how different parameter choices affect the pseudoenergy landscape of RuvB's cycle, showing a small impact on the overall trend S9 Fig.

**Computing polar interactions.** We identified the polar interactions of each atom using a script adapted from pymol [87] core functions, `show_contacts()`. We computed all inter and intramolecular H-bonds, optimal polar interactions (cutoff 3.5 Å) and suboptimal polar interactions (cutoff 3.9 Å). To calculate the total elastic pseudoenergy of intermolecular interactions, we sum the elastic pseudoenergy of all interacting residues weighted by the fraction of such interactions established with that particular molecule, see S5 Fig for an example.

**Stochastic kinetic model of RuvB assembly.** Consider a system with $M$ molecules, labelled $m \in \{1, 2, \dots, M\}$, representing each of the RuvB subunits. These molecules populate a discrete and ordered set of states, representing the protein conformations, $i \in \{1, 2, \dots, N\}$. For the RuvB cycle, $M = 6$ and $N = 30$. Transitions in this system are caused by random interactions with a thermal reservoir of temperature $T$, inducing a molecule $m$ in state $i$ to change to one of the neighbour states ($i + 1$ or $i - 1$). We set periodic boundary conditions, allowing transitions between states 1 and $N$. We assume that a transition changes the state of a single molecule at a time.

Each of the $N$ states is associated with a reaction coordinate $\theta(i) = 2\pi i/N$. To simplify our notation, we denote as $\theta_m$ the reaction coordinate of the subunit $m$. In our analyses, we

assume that these reaction coordinates correspond to the conformations observed in RuvB's mechanochemical cycle, $\theta(1) \to$ F1, $\theta(2) \to$ F2, ..., $\theta(30) \to$ A5. Thus, a hexameric assembly state can be fully described by a vector with the reaction coordinates of each subunit, for instance, $\mathbf{a} = [\theta_1(4), \theta_2(9), \theta_3(14), \theta_4(18), \theta_5(24), \theta_6(29)]$ corresponds to the assembly shown in S10 Fig.

The internal energy of the whole assembly can be decomposed in two terms. First, each state is characterised by a potential elastic energy $\Delta E_{el}(\theta)$. Second, a coupling energy $E_c$ links pairs of molecules with consecutive indices, see Fig 7A. We define the coupling energy inspired by the Kuramoto model [88], a classical mathematical model for describing coupled phase oscillators:

$$E_c(\theta_m, \theta_{m+1}) = \frac{K}{M}\left(1 - \cos(\theta_{m+1} - \theta_m - \delta)\right),$$ 

(5)

where $K$ and $\delta$ respectively represent the coupling strength and the phase-lag in reaction coordinate between consecutive molecules at equilibrium. Because we are interested in modelling RuvB, which operates as a sequential motor, we fix $\delta = \pi/3$. This ensures that the five possible arrangements in which subunits are evenly spaced in conformation–matching the states $s_1, s_2, ..., s_5$–are minima of $E_c$. The energy of the assembly can be thus summarised as

$$E(\theta) = \sum_{m=1}^{M}\left[\Delta E_{el}(\theta_m) + E_c(\theta_m, \theta_{m+1})\right].$$

(6)

The energetics and dynamics of the system are connected by local detailed-balance (S10 Fig), which links transition rates between pairs of states and their respective energy difference [20]. As we mentioned previously, transitions are allowed between assembly states for which one of the molecules $m$ changes to a neighbouring state. From the allowed transitions, a few have an additional energetic contribution $\Delta\mu$, representing the external chemical drive coming from the ATP cycle. We set the kinetic rates between two arbitrary neighbouring states $\mathbf{a} = \theta$ and $\mathbf{b} = \theta'$ as such:

$$k_{ab} = \begin{cases} \tau_{ab}^{-1}\Gamma_{ab} & \text{if } \mathbf{a} \leftrightarrow \mathbf{b} \text{ is not driven} \\ \tau_{ab}^{-1}\Gamma_{ab}\exp\left(\frac{\Delta\mu}{2k_BT}\right) & \text{if } \mathbf{a} \to \mathbf{b} \text{ is driven} \\ \tau_{ab}^{-1}\Gamma_{ab}\exp\left(\frac{-\Delta\mu}{2k_BT}\right) & \text{if } \mathbf{a} \leftarrow \mathbf{b} \text{ is driven,} \end{cases}$$

(7)

with $\Gamma_{ab} = \left(1 + \exp((E(\mathbf{b}) - E(\mathbf{a}))/k_BT)\right)^{-1}$, $k_B$ the Boltzmann constant, and $\tau_{ab}$ the characteristic time scale of the transition (note that $\tau_{ab} = \tau_{ba}$). As the local detailed-balance condition imposes a constraint only in the ratio of rates, the form we set for $k_{ab}$ is one of many possible choices. See Appendix F in S1 Text for a complete description of how $\tau_{ab}$ is defined and calibrated.

We simulated stochastic trajectories based on this kinetic model using the Gillespie algorithm [89].

## Supplementary information

**S1 Text. Supplementary information.**
(PDF)

**S1 Scripts. Scripts containing all the analyses.**

**S1 Fig. The structure $s_0$ in perspective of the conformational cycle.** Multidimensional scaling analysis with cycle structures and the initial state $s_0$ (mean of 100 MDS) using **A.** The mean largest eigenvalue of the strain tensor, $\langle \epsilon_3 \rangle$, **B.** The mean lowest eigenvalue of the strain tensor, $\langle -\epsilon_1 \rangle$, and **C.** The total elastic pseudoenergy $\Delta E_{el}$ as dissimilarity measures. Despite most of the $s_0$ conformations lying before $s_1$ in clockwise order in this representation, the conformations in blue (corresponding to position C) and green (position D) are in different locations in MDS coordinates. That can be rationalised by the absence of interaction with RuvA in conformation C0 and the different nucleotide composition in position D (which harbours an ATP in conformation D0). The proximity of $s_0$ and $s_1$ for positions A, F and E suggests that $s_0$ precedes $s_1$ in order. However, the nucleotide composition of $s_0$ subunits does not accommodate it as part of the mechanochemical cycle, suggesting that a priming ATP hydrolysis in D0 is required to start the mechanochemical cycle without a position switch. Note that in the cycle conformations, hydrolysis happens only in position A, as opposed to D. Therefore, considering $s_0$ as an initial state not yet engaged in the mechanochemical cycle is a parsimonious interpretation of such MDS analyses.
(TIFF)

**S2 Fig. Statistics of RuvB energetics and volume. A.** Histogram of elastic pseudoenergies per residues in all thirty monomeric conformations. The elastic pseudoenergy distribution, depicted in log scale, shows that $\Delta E_{el} < 0.01 k_B T$ for most residues in all conformations, with heavier tails for conformations in positions E and D. **B.** Total pseudoenergy distribution among assemblies is less variable than the pseudoenergy distribution among positions. **C.** Volume per conformation displays very low variation (around 0.1%) for the van der Waals volume, which was considered for the elastic pseudoenergy calculations.
(TIFF)

**S3 Fig. Elastic pseudoenergy profiles and intermolecular interactions performed by the mechanically active regions. A.** Mean and standard deviation of elastic pseudoenergy of residue trajectories are highly correlated. The dashed line defines a threshold on variance, selecting the top 10% more elastically variable residues. **B.** Correlation between elastic pseudoenergy profiles reveals clusters of residues with similar trajectories. The hierarchical clustering is based on the Euclidean distance among correlations. **C.** Mapping of spatial localisation of mechanically active regions. **D.** Elastic pseudoenergy profile of regions 1 to 4 and their specific intermolecular interactions.
(TIFF)

**S4 Fig. Identification of regions of interest based on elastic pseudoenergy and $\beta$-factor. A.** Approach used in the main text to identify regions of mechanical activity considering the top-10% residues on the standard deviation of elastic pseudoenergy. **B.** Regions of high $\beta$-factor standard deviation (top 10%). Residues marked in red agree with the elastic pseudoenergy criterion, whereas those in blue are unique to the $\beta$-factors standard deviation threshold. Residues identified above the threshold are highlighted in red and shown in a projection of RuvB monomer (below). Residues with high variability on $\beta$-factors are enriched with residues from Region 3 **C.** Regions of high $\beta$-factor means (top 10%). Residues with high mean $\beta$-factors are enriched with residues from Region 4. The results show that using $\beta$-factors does not fully recover the regions of interest identified through elastic pseudoenergy analysis, suggesting they should be interpreted as complementary rather than redundant metrics.
(TIFF)

**S5 Fig. Polar interactions. A.** Example of polar interactions identified for R21 (in mechanically active region 1), which displays interactions in *cis*, with the forward RuvB subunit, and with the nucleotide. The elastic pseudoenergy of interaction $\Delta E_{inter}$ is calculated as the average of the elastic pseudoenergy of the residue weighted by the number $n_i$ of interactions performed. **B.** Number of polar interactions per state grouped by type of interaction. Few residues do not display any of the considered types of polar interactions. **C.** Total elastic pseudoenergy with intra and intermolecular interactions and its associated statistics. We considered only the states where there was at least one interaction. On the right, we show the statistics tested between pairs of groups (Mann-Whitney U test, * corresponds to $p < 0.05$, Bonferroni corrected).
(TIFF)

**S6 Fig. Elastic pseudoenergy and DNA interaction profiles.** Elastic pseudoenergy quantification for residues that interact with the DNA via polar interactions in at least one conformation. Circle markers indicate states with predicted interactions, and the dashed line represents RuvB's median elastic pseudoenergy profile.
(TIFF)

**S7 Fig. Elastic pseudoenergy content and secondary structure are correlated in Sensor 1 motif. A.** The residues highlighted in magenta represent part of the mechanically active region 2, found by our approach. Conformations from F to D, except for F1 and F3, were predicted to have a loop as a secondary structure, whereas conformations from C to A display a transient $3_{10}$ helix. **B.** The total pseudoenergy of residues R160, A161, and G162 is shown according with the secondary structured. The total pseudoenergy of these residues when found on loop conformation is significantly different than when on $3_{10}$ helix conformation (Mann-Whitney U test, $p < 0.05$).
(TIFF)

**S8 Fig. Relationship between elastic pseudoenergy and sequence conservation. A.** Map of amino acid conservation among RuvBs from 18 bacterial species [90] and the construct we analysed. The conserved motifs correspond to regions previously assigned by [90], to which the extent of conservation ranges from RuvB-specific to P-loop NTPases. **B.** Sequence conservation is not correlated with the mean elastic pseudoenergy per residue (Kruskal-Wallis H-test, $p > 0.05$). **C.** Mean elastic pseudoenergy of residues belonging to different conserved motifs. We found that the groups are significantly different (Kruskal-Wallis H-test, $p < 0.05$), but the pattern is not consistent across the different motifs. Motifs 2 and 9 were less energetic than non-conserved residues, whereas Motifs 4 and 6 were more energetic. Note that Walkers A and B (motifs 3 and 5) are not significantly more or less strained than non-conserved regions. (Mann-Whitney U-test for each motif *vs* other residues, Bonferroni corrected $p < 0.05$).
(TIFF)

**S9 Fig. RuvB elastic pseudoenergy profile is robust to different parameter choices.** We systematically repeated the calculations for elastic pseudoenergy at different states, changing the atoms selected for the analysis, the radius of the local neighbourhood, and the method to assign weights to the atoms in the neighbourhood. We show the Pearson's correlation ($\rho$) between the pseudoenergy landscapes obtained with different parameter choices and the main text analysis (backbone atoms, $r = 9$Å, intersect all structures). Overall, the main text parameter choice agrees qualitatively and quantitatively with other choices being, in general, more conservative (typically is as deformed as or less deformed than other pseudoenergy landscapes). All profiles display a similar trend, except for alpha carbons and radius $r = 9$Å

due to imprecise calculations when the number of atoms in the local neighbourhood is small. **A/B/C.** elastic pseudoenergy profiles change the radius of the neighbourhood and the atom selection criteria. For these tests, we use the same weighting method from the main text. **D.** elastic pseudoenergy profile for different weight method, using backbone atoms and radius to 9Å. The Venn diagrams represent the atoms in the neighbourhood of radius $r$ for different structures $s_i$, for the example $s_1$ vs $s_0$.
(TIFF)

**S10 Fig. Kinetic model formulation. A.** Representation of a particular transition between assembly states. A RuvB assembly consists of six coupled subunits, represented by circles connected through springs. In a given hexameric state, each subunit occupies one of the 30 conformations of the mechanochemical cycle, see tables below schematics for the example states **a** and **b**. Transitions among adjacent states are controlled by the kinetic rates $k$. In the example, the green circle transitions between the adjacent conformations C3 and C4, where the other subunits remain in the same configuration. Red arrows highlight the transitions driven by the chemical potential $\Delta\mu$, which are associated with ADP release and ATP binding **B.** The detailed-balance condition enforces the ratio of back-and-forth transition rates to respect the energy difference of the assembly configurations. The driven transitions, however, are biased towards one direction.
(TIFF)

**S1 Video. Localisation and interactions of the mechanically active region 1.** The panels display the backbone of RuvB subunit and the adjacent forward subunit, with which region 1 interacts in trans. The alpha carbons of residues in region 1 are highlighted in dark purple and the residues that interact with region 1 through polar contacts are in light purple.
(GIF)

**S2 Video. Localisation and interactions of the mechanically active region 2.** The panels display the backbone of RuvB subunit and the adjacent backward subunit, with which region 2 interacts in trans. The alpha carbons of residues in region 2 are highlighted in dark pink and the residues that interact with region 2 through polar contacts are in light pink.
(GIF)

**S3 Video. Localisation and interactions of the mechanically active region 3.** The panels display the backbone of RuvB subunit and the adjacent forward subunit, with which region 3 interacts in trans. The alpha carbons of residues in region 3 are highlighted in dark green and the residues that interact with region 3 through polar contacts are in light green.
(GIF)

**S4 Video. Localisation and interactions of the mechanically active region 4.** The panels display the backbone of RuvB subunit and the adjacent backward subunit, with which region 4 interacts in trans. The alpha carbons of residues in region 4 are highlighted in dark orange and the residues that interact with region 4 through polar contacts are in light orange.
(GIF)

## Acknowledgments

P.S. and V.H.M. would like to thank J. Howard for illuminating discussions. V.H.M thanks M. Machuqueiro for insightful discussions on molecular dynamics force-fields. All authors thank W. Lugmayr for discussions concerning PSA implementation.

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
