## [Decision Letter · Decision Letter 0]

30 Sep 2025

PCOMPBIOL-D-25-01737

Elastic analysis bridges structure and dynamics of an AAA+ molecular motor

PLOS Computational Biology

Dear Dr. Mello,

Thank you for submitting your manuscript to PLOS Computational Biology. After careful consideration, we feel that it has merit but does not fully meet PLOS Computational Biology's publication criteria as it currently stands. Therefore, we invite you to submit a revised version of the manuscript that addresses the points raised during the review process.

Please submit your revised manuscript within 30 days Nov 30 2025 11:59PM. If you will need more time than this to complete your revisions, please reply to this message or contact the journal office at ploscompbiol@plos.org. Please include the following items when submitting your revised manuscript:

We look forward to receiving your revised manuscript.

Kind regards,

Alexander MacKerell

Academic Editor

PLOS Computational Biology

Arne Elofsson

Section Editor

PLOS Computational Biology

**Journal Requirements:**

At this stage, the following Authors/Authors require contributions: Victor Hugo Mello, Jiri Wald, Thomas C Marlovits, and Pablo Sartori. Please ensure that the full contributions of each author are acknowledged in the "Add/Edit/Remove Authors" section of our submission form.

5) We have noticed that you have uploaded Supporting Information files, but you have not included a list of legends. Please add a full list of legends for your Supporting Information files after the references list.

Potential Copyright Issues:

- Figure 4F. Please confirm whether you drew the images / clip-art within the figure panels by hand. If you did not draw the images, please provide (a) a link to the source of the images or icons and their license / terms of use; or (b) written permission from the copyright holder to publish the images or icons under our CC BY 4.0 license. Alternatively, you may replace the images with open source alternatives. See these open source resources you may use to replace images / clip-art:

- Figure 4A, Region_1, 2, 3, and 4. Please (a) provide a direct link to the base layer of the map (i.e., the country or region border shape) and ensure this is also included in the figure legend; and (b) provide a link to the terms of use / license information for the base layer image or shapefile. We cannot publish proprietary or copyrighted maps (e.g. Google Maps, Mapquest) and the terms of use for your map base layer must be compatible with our CC BY 4.0 license.

7) Please ensure that the funders and grant numbers match between the Financial Disclosure field and the Funding Information tab in your submission form. Note that the funders must be provided in the same order in both places as well.

**Reviewers' comments:**

Reviewer's Responses to Questions

**Comments to the Authors:**

Reviewer #1: In this manuscript, the authors present a detailed application of the innovative concept of “pseudoenergies” that they introduced previously. As their protein of choice, they focused on the conformational changes of RuvB during its hand-over-hand cycle along DNA. The paper has two parts: one is a sophisticated data analysis of the structural changes and associated energies, and the other is a kinetic non-equilibrium model of its function. The model partly incorporates insights from the first part, but makes non-trivial predictions that are cross-validated. The paper's conclusions are substantially supported by the supplementary material, and the limitations are clearly discussed.

This study is an original and substantial contribution to the open question of how to connect protein structure to function. Therefore, I recommend publication in PLOS CB

I have a few recommendations for the authors:

It is essential to note that it is fundamentally impossible to obtain kinetic information from static measurements, such as structures. Therefore, I was first very confused when I read the sentence in line 138. For me, it somehow implies that the cycle can be determined from ordering the conformations according to a principle of minimal elastic deformation. For a non-equilibrium system that cycles through a pathway of discrete states, the energy barriers (not the energy differences) between the states dictate the dynamics. It would be helpful if the authors could clarify that the elastic pseudoenergies can only contribute to equilibrium considerations, and insights into non-equilibrium phenomena must come from dynamic measurements. Therefore, it is surprising to me that the minimal elastic deformation principle suggests a reasonable sequence of states. It could also be a general principle in biology that I am not aware of. It would be interesting to discuss if there is a case of a sequence of conformational states that cannot be revealed from the minimum pseudoenergy principle. Perhaps it is possible to construct a simple example in which the energy barriers are chosen such that the cycle does not follow the minimum difference in internal energies between the states.

It is worth noting that the factor 2 in Eq. (7) is a choice, because detailed balance also holds for other ways of distributing “the driving” either to the forward or the backward rate.

I would like to express my respect to the authors for such a detailed analysis and clear presentation. Nice work!

Reviewer #2: Attached

**Have the authors made all data and (if applicable) computational code underlying the findings in their manuscript fully available?**

Reviewer #1: Yes

Reviewer #2: Yes

PLOS authors have the option to publish the peer review history of their article (what does this mean?). If published, this will include your full peer review and any attached files.

Reviewer #1: No

Reviewer #2: **Yes: **Christian S. Parry

**Figure resubmission:**
---

## [Editor Report · Decision Letter 1]

7 Oct 2025

Dear Mello,

We are pleased to inform you that your manuscript 'Elastic analysis bridges structure and dynamics of an AAA+ molecular motor' has been provisionally accepted for publication in PLOS Computational Biology.

Best regards,

Alexander MacKerell

Academic Editor

PLOS Computational Biology

Arne Elofsson

Section Editor

PLOS Computational Biology

---

## [Editor Report · Acceptance letter]

PCOMPBIOL-D-25-01737R1

Elastic analysis bridges structure and dynamics of an AAA+ molecular motor

Dear Dr Sartori,

I am pleased to inform you that your manuscript has been formally accepted for publication in PLOS Computational Biology. Your manuscript is now with our production department and you will be notified of the publication date in due course.

With kind regards,

Judit Kozma
